# SparseInteraction: Sparse Semantic Guidance for Radar and Camera 3D Object Detection

Shaoqing Xu*
University of Macau
Macau, China
shaoqing.xu@connect.um.edu.mo

Shengyin Jiang*
Beijing University of Posts
and Telecommunications
Beijing, China
shengyin@bupt.edu.cn

Fang Li*
Beijing Institute of
Technology
Beijing, China
lifang980825@gmail.com

Li Liu
Beijing Institute of
Technology
Beijing, China
liuli.ll9412@gmail.com

Ziying Song
Beijing Jiaotong University
Beijing, China
songziying@bjtu.edu.cn

Bo Yang
Beijing University of Posts
and Telecommunications
Beijing, China
bobyang@bupt.edu.cn

Zhi-xin Yang†
University of Macau
Macau, China
zxyang@um.edu.mo

## Abstract

Multi-modal fusion techniques, such as radar and images, enable a complementary and cost-effective perception of the surrounding environment regardless of lighting and weather conditions. However, existing fusion methods for surround-view images and radar are challenged by the inherent noise and positional ambiguity of radar, which leads to significant performance losses. To address this limitation effectively, our paper presents a robust, end-to-end fusion framework dubbed SparseInteraction. First, we introduce the Noisy Radar Filter (*NRF*) module to extract foreground features by creatively using queried semantic features from the image to filter out noisy radar features. Furthermore, we implement the Sparse Cross-Attention Encoder (*SCAE*) to effectively blend foreground radar features and image features to address positional ambiguity issues at a sparse level. Ultimately, to facilitate model convergence and performance, the foreground prior queries containing position information of the foreground radar are concatenated with predefined queries and fed into the subsequent transformer-based decoder. The experimental results demonstrate that the proposed fusion strategies markedly enhance detection performance and achieve new state-of-the-art results on the nuScenes benchmark. Source code is available at https://github.com/GG-Bonds/SparseInteraction.

## CCS Concepts

• **Computing methodologies → Object detection**.

## Keywords

3D Object Detection, autonomous driving, Multi-modal

*Equal contribution
†Corresponding author.

**ACM Reference Format:**
Shaoqing Xu, Shengyin Jiang, Fang Li, Li Liu, Ziying Song, Bo Yang, and Zhi-xin Yang. 2024. SparseInteraction: Sparse Semantic Guidance for Radar and Camera 3D Object Detection. In *Proceedings of the 32nd ACM International Conference on Multimedia (MM '24), October 28-November 1, 2024, Melbourne, VIC, Australia.* ACM, New York, NY, USA, 10 pages. https://doi.org/10.1145/3664647.3681565

## 1 Introduction

Perception of 3D obstacles via different types of sensors is a fundamental task in the field of computer vision and robotics. The fusion of LiDAR and camera technologies has achieved high accuracy performance [19, 26, 28, 29, 37–39]. However, cameras face limitations in low-visibility conditions like heavy fog or rain while the high cost of LiDAR poses further challenges.

Radar sensors are notable for their affordability, resilience in all weather conditions and capacity for accurate speed estimations over considerable distances [3, 20, 30, 32, 33, 41]. Nevertheless, the direct usability of radar points is impeded by the inherent challenges of noise and positional ambiguity.

Recent studies achieved convincing performance by transforming image features onto the BEV (Bird's Eye View) and fusing them with radar features. In particular, CRN [11] fuses image-bev-features and radar features based on transformer architecture to resolve the ambiguity in radar positioning. However, the method requires assistance from LiDAR point clouds during the training process. Additionally, the presence of noisy radar data before the fusion process leads to suboptimal fusion features. For filtering the noisy information of radar, the CramNet [8], depicted in Figure 1-(a) directly eliminates noisy radar by leveraging radar features. Conversely, the approach illustrated in Figure 1-(b), such as HVDetFusion [12], utilizes detection bounding boxes from the image to filter noisy information. Additionally, methods(e.g., MVFusion [36]) like in Figure 1-(c), incorporate foreground classification from the image to filter irrelevant information by projecting 3D radar points onto 2D image pixels. However, a common limitation of these methods is their reliance on the characteristics of a single modality before filtering. This singular focus tends to overlook the potential benefits of integrating multiple modalities which leads to suboptimal outcomes.

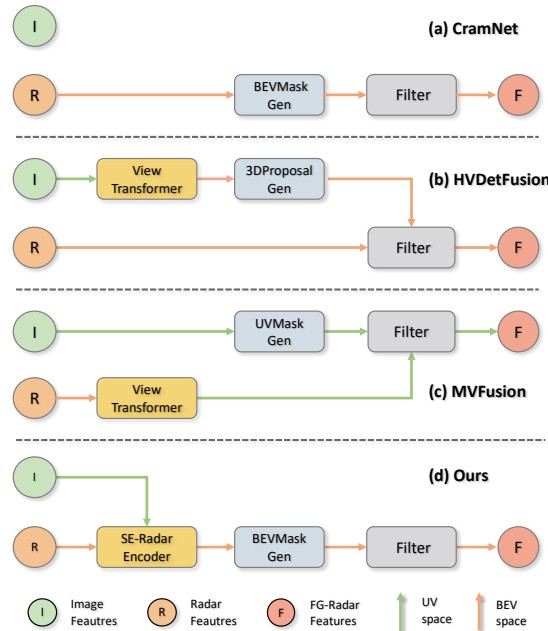

**Figure 1: Comparing different design pipelines for filtering noisy radar. (a) CramNet, using radar features to generate filtering conditions, is constrained by the limited representational capacity of single-modality features. (b) HVDetFusion, translating image features into BEV space to create 3D proposals as filtering conditions, is effective but complex. (c) MVFusion employs UV space mask prediction as a filtering conditions, facing challenges in depth differentiation at identical image locations. (d) Our approach initially enhances radar features with semantic image information via the Semantic-enhanced Radar Encoder (SRE), followed by the effective filtering of noisy radar in BEV space.**

In this paper, we present a robust, transformer-based, end-to-end fusion framework named **SparseInteraction**. Firstly, considering the negative impact of noisy radar on feature fusion, we introduce the Noisy Radar Filter module composed of Semantic-enhanced Radar Encoder (**SRE**) and Foreground Radar Mask (**FRM**) to preserve useful radar information under the guidance of image semantic information. Subsequently, we leverage the sparse foreground radar features and their related effective image feature areas to enhance the interactive information of multimodal features through Sparse Cross-Attention Encoder(**SCAE**) which effectively addresses feature misalignment. This fusion method not only resolves the issue of radar's ambiguous positioning but also provides robust results even in the event of sensor failures. Furthermore, we extract foreground object queries from both foreground radar features and enhanced BEVusion features in an efficient way. Finally, the foreground prior queries containing position information of the foreground radar are concatenated with predefined queries and fed into the subsequent transformer-based decoder, for facilitating the model convergence and performance. Ultimately, our framework

achieves impressive and robust 3D object detection results without LiDAR points cloud in either the training or testing phases.

In general, the contribution of this work can be summarized as follows:

- We introduce a novel Noisy Radar Filter module designed to efficiently address false positive issues by filtering out useless information under the guidance of image semantic features and radar features.
- To further effectively address the challenge of radar positioning ambiguity, an innovative sparse query-based module, Sparse Cross-Attention Encoder, is proposed for featuring a multi-modal representational interaction. Notably, it incorporates high-quality radar 3D object priors into 3D adaptive queries.
- Experiments of our framework SparseInteraction on the nuScenes dataset achieve state-of-the-art performance based on transformer technology which is independent of LiDAR data. The proposed foreground radar filtering is expected to spur future research.

## 2 Related Work

### 2.1 Camera Based 3D Object Detection

Vision-based 3D object detection is a promising perception task in autonomous driving. Recently, BEV(Bird's Eye View) based 3D object detection methods caught more eye in the academic world. This approach can be categorized into bottom-up methods, such as LSS [24], BEVDet [7], and BEVDepth [14], which transform image features into BEV features through depth prediction. Conversely, top-down methods pre-define queries in 3D space and project onto images for sampling. For instance, DETR3D [35] uses learnable 3D queries for end-to-end detection pipelines without NMS post-processing. BEVFormer [15] generates BEV features by taking grid-like queries to sample relevant features from images.

### 2.2 Multi-modal 3D Object Detection

While FUTR3D [2] employs 3D reference points as queries to sample features from the projected view and BEVFusion [19] uses a lift-splat-shoot operation to project image features onto BEV space and then fuse them with LiDAR features.

Similar to LiDAR, radar can offer potential position and precision velocity information. However, challenges often arise due to the inherently noisier, sparser points and vague height information while cost-effectiveness and functionality in various conditions make it an appealing option for enhancing camera robustness [27]. CRN [11] transforms image feature to BEV space and integrates them with radar BEV features via a transformer structure while requiring LiDAR data for depth supervision. To mitigate radar noise, some studies first process the foreground radar, which results in more significant outcomes. e.g., CramNet [8] and HVDetFusion [12]. However, these methods do not integrate features from the other modality before extracting foreground information, leading to sub-optimal extraction and complexity.

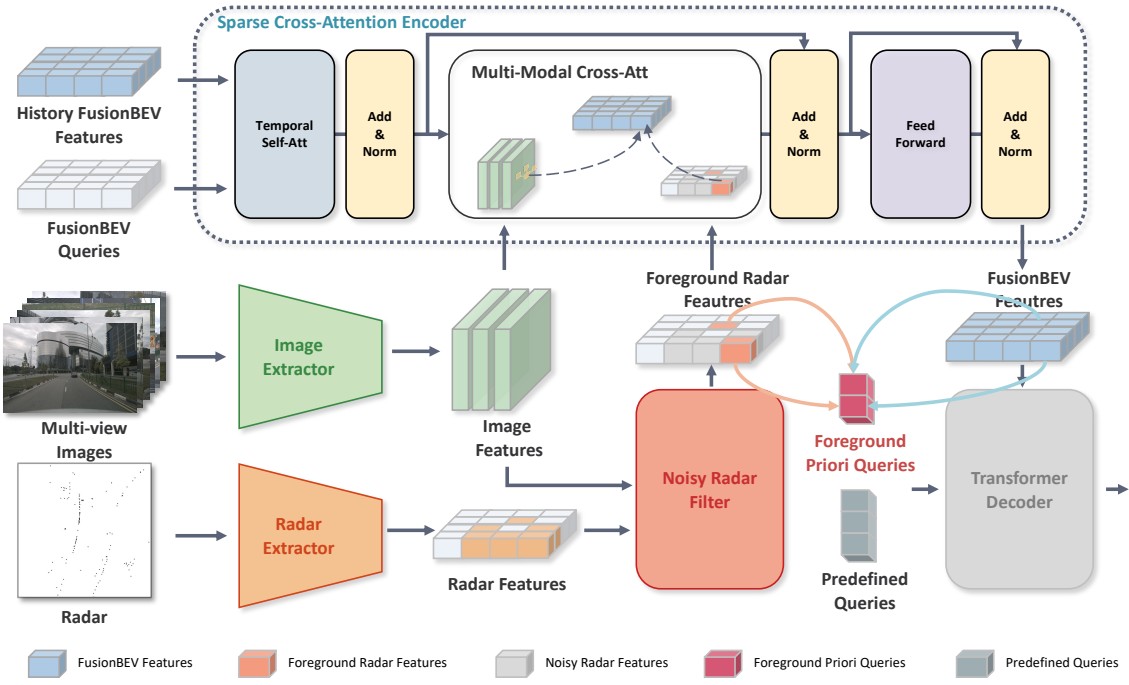

**Figure 2: The overall architecture of the proposed SparseInteraction.**

## 3 Methodology

An overview of the proposed framework **SparseInteraction** is illustrated in Figure 2. We first extract features from radar points cloud and multi-view images, respectively. To address noisy radar information, the Noisy Radar Filter is proposed to filter useless features by combining both radar features and high-level semantic features of images. Then, the enhanced foreground radar features and image features are sent into the Sparse Cross-Attention Fusion Encoder, which utilizes deformable attention mechanism [44]. Compared to previous approaches such as BEVFusion [19], our Sparse Cross-Attention Encoder effectively resolves radar localization ambiguity without LiDAR depth supervision. Finally, prior information from the foreground radar features is encoded as foreground prior queries concatenated with predefined queries and introduced as object queries in transformer decoder layers.

### 3.1 Modal Features Extractor

*3.1.1 **Image Feature Extractor.*** Following BEVFormer [15], we adopt ResNet101 [5] as the backbone for extracting image features from multi-view images and use a standard Feature Pyramid Network (FPN) [17] on top of the backbone to generate multi-scale image feature representations $F_c^i, i \in [1, 2, \cdots, N]$, where $i$ denotes each view of cameras.

*3.1.2 **Radar Feature Extractor.*** Given the limitation of radar in providing reliable elevation, we utilize pillar-based representation for radar points cloud. Following FUTR3D [2], we adopt Pillar-Net [25] and multi-layer perception (MLP) to extract radar features

in BEV space, denoted as $F_r \in \mathbb{R}^{C \times H \times W}$, where $C$ is the dimension of radar features and $(H, W)$ represents the BEV resolution.

### 3.2 Noisy Radar Filter

Despite radar recording the velocity and location of objects, it also inadvertently captures information of non-target which leads to false positive detection issues. To overcome this limitation, we propose an innovative lightweight module Noisy Radar Filter (NRF), comprising two key components: *Semantic-enhanced Radar Encoder* and *Foreground Radar Mask*. The detailed architecture of **NRF** module is depicted in Figure 3.

*3.2.1 **Semantic-enhanced Radar Encoder.*** To migrate noisy radar information, we take non-empty features in $F_r$ as queries for aggregating neighboring features guided by semantic information from the image. More importantly, a substantial portion of features originating from the radar encoder is vacant owing to the inherent sparsity of radar points. Consequently, we employ a 3x3 bias-free convolution layer to disperse foreground radar features into vacant positions, thereby augmenting the richness of radar features. Subsequently, the valid radar features are onto corresponding image features to generate semantic-enhanced radar features, $F_r^s \in \mathbb{R}^{C \times H \times W}$. Specifically, considering radar's uncertain height information, for a valid radar features pixel $Q_{x,y}^r$ located at $(x, y)$, we lift it to $N_{ref}$ 3D points with different heights $z_j$ and projecting them onto the $i$-th image at projection pixels $(u_i, v_i)$. Then valid radar features are enhanced using deformable attention [44] with the image features $F_c^i$. The entire enhanced process is represented

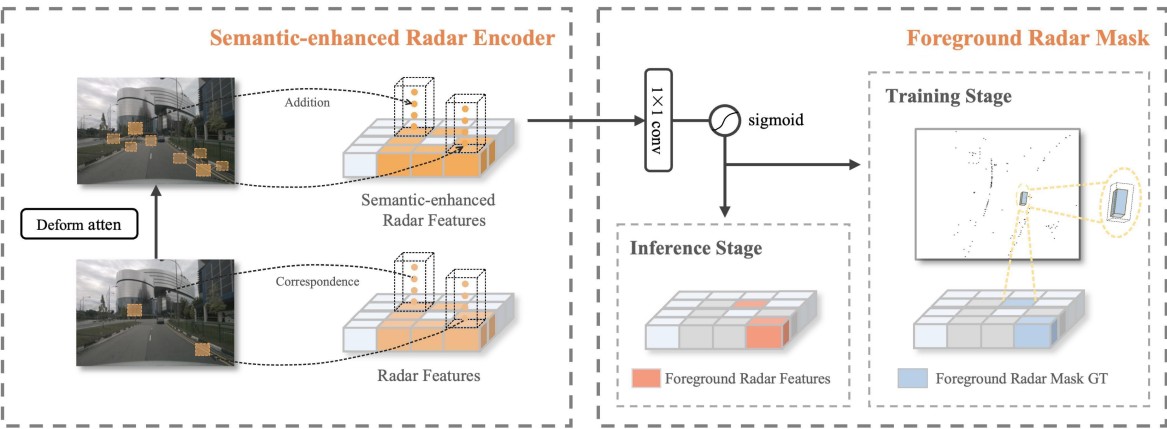

**Figure 3: The architecture of proposed Noisy Radar Filter module is designed to filter noisy features of radar. Taking radar and camera features as input effectively filters out noise under the guidance of high-level semantic information from the image.**

as follows:

$$Q_{x,y}^s = \sum_{i=1}^{N_c} \sum_{j=1}^{N_{ref}} DefAttn(Q_{x,y}^r, P_i(x,y,z_j), F_c^i) \quad (1)$$

where $Q_{x,y}^s$ is the semantic-enhanced radar features pixel located at $(x, y)$. $P_i(x, y, z_j)$ represents projection point of 3D point $(x, y, z_j)$ in the image coordinate system of the $i$-th camera.

*3.2.2* ***Foreground Radar Mask.*** Acknowledging that radar points predominantly manifest near objects, only a few points exist within 3D bounding boxes due to the ambiguity in radar localization. We figured that radar points in proximity to 3D bounding boxes also contribute to the detection process. Therefore, we enlarge the 3D bounding boxes derived from ground truth labels to 1.5 times of original dimensions, projecting them onto the BEV view as binary mask ground truth $M_{gt}$. We employ a 1x1 convolution layer paired with a sigmoid activation function to classify the semantic-enhanced radar features generated above. To ensure a superior recall, a lower foreground threshold $\gamma$ is used. Consequently, radar features with foreground score $s_i$ exceeding the threshold $\gamma$ are classified as foreground radar features $F_r^f$; the reverse applies for lower scores as background radar features $F_r^b$.

## 3.3 Sparse Cross-Attention Encoder.

While CRN [11] first generates sparse and discrete BEV features of the image under the LiDAR depth supervision [16] and fuses with radar features through the attention mechanisms which leads to suboptimal fusion results. To address this, we propose the Sparse Cross-Attention Encoder (SCAE), a novel Multi-modal fusion module based on transformers. As shown in Figure 2, SCAE comprises six encoding layers, each consisting of Temporal Self-Attention (**TSA**) and Multi-modal Cross-Attention (**MCA**). First, a set of learnable FusionBEV queries interact with historical FusionBEV features through TSA. Specially, for optimizing computation and more precision information, only sparse fusion features of the foreground are utilized as valid queries in the subsequent module. To acquire more

comprehensive feature information, the relevant key values from the image feature map and foreground radar features (generated by Section 3.2) will be queried in the MCA submodule. Subsequently, the FusionBEV queries are updated by a feedforward network as input for the next layer. After six encoding layers, we generate precise and robust FusionBEV features. Significantly, all the operations of our SCAE module are in sparse space which requires less computation compared to FusionFormer [6].

*3.3.1* ***FusionBEV Queries.*** We defined a set of learnable parameters $Q \in \mathbb{R}^{C \times W \times H}$ called FusionBEV queries. Here, $W \times H$ represents the resolution of BEV view, and $C$ is the channel of FusionBEV queries. These queries are shared across all modalities, directly fusing images features $F_c$ and foreground radar features $F_r^f$.

*3.3.2* ***Temporal Self-Attention.*** Temporal information plays a crucial role in 3D object detection. Following BEVFormer [15], Initially, we align historical FusionBEV features based on ego-motion. Subsequently, we employ Temporal Self-Attention to effectively fuse these historical FusionBEV features. The process is as follows expression:

$$TSA\left(Q_p, \left(Q, B'_{t-1}\right)\right) = \sum_{V \in \{Q, B'_{t-1}\}} DefAttn\left(Q_p, p, V\right). \quad (2)$$

where $Q_p$ denotes the FusionBEV query located at $p = (x, y)$. $B'_{t-1}$ represents the FusionBEV features at timestamp $t - 1$ after temporal alignment.

*3.3.3* ***Multi-modal Cross-Attention.*** As detailed in Section 3.2, the positions of the background have been established in the BEV space. To optimize computational efficiency, FusionBEV queries in these positions are excluded from the computation, which makes FusionBEV queries focus on the potential target object positions $p_t = (x_t, y_t)$. More importantly, these queries are directly projected onto each modality feature map and enhanced with deformable attention [44], eliminating the need to transform image features into BEV space beforehand. This process is represented by the

following formula:

$$MCA\left(Q_{p_t}, F_c, F_r^f\right) = DefAttn\left(Q_{p_t}, P_{2D}\left(p_t\right), F_r^f\right)$$
$$+ \sum_{i=1}^{N_c} \sum_{j=1}^{N_{ref}} DefAttn\left(Q_{p_t}, P_{3D}^i\left(p_t, z_j\right), F_c^i\right) \quad (3)$$

where $Q_{p_t}$ denotes the FusionBEV query located at $p_t$. $P_{2D}(p_t)$ denotes the projected positions on the BEV space. As FusionBEV queries lack height information, we employed the method similar to described in Section 3.2, which lifting points $p_t$ with different heights $z_j$ to obtain $N_{ref}$ 3D projection points $(x_p, y_p, z_j)$, where $P_{3D}^i(x_p, y_p, z_j)$ are the corresponding positions of these 3D projection points on image $F_c^i$.

## 3.4 Foreground Prior Queries

Our 3D detection head adopts the decoder from BEVFormer [15], similar to the decoder in Deformable DETR [44]. It is well known that DETR-style predefined queries struggle to converge due to the lack of prior knowledge [22, 40]. Inspired by two-stage methods [40], we encode the prior information of foreground radar features near the region of interest directly into the decoder. To minimize the effect of low-quality foreground radar, a higher threshold $r$ is adopted, ensuring only the most salient foreground radar features are considered. Each foreground radar feature pixel in the BEV plane only provides 2D position $p_f = \left(x_f, y_f\right)$, and the approximate height $z_f^{pred}$ of each foreground radar feature pixel is predicted using the FusionBEV features $F_b$ at the specific location $p_f$, expressed as:

$$z_f^{pred} = sigmoid\left(MLP(F_b^f)\right) \quad (4)$$

Where $F_b^f$ is sampled feature from FusionBEV features at position $p_f$.

After obtaining the foreground prior 3D positional information $(x_f, y_f, z_f^{pred})$, we encode it into foreground prior queries as follows:

$$Q_{sem} = SemEncoder\left(F_b^f, s_i\right) \quad (5)$$

$$Q_{pos} = PosEncoder\left(x_f, y_f, z_f^{pred}\right) \quad (6)$$

$$Q_{prior} = Q_{sem} + Q_{pos} \quad (7)$$

where $s_i$ represents the confidence score of the foreground radar as predicted in Section 3.2. $Q_{sem}$ and $Q_{pos}$ represent the semantic feature embedding and position embedding, respectively. $PosEncoder$ consists of a sinusoidal transformation [31], and another MLP, with $SemEncoder$ also being an MLP. And $Q_{prior}$ represents the final foreground prior queries.

Finally, the encoded foreground prior queries and predefined queries are concatenated and then fed into the decoder.

## 3.5 Head and Loss

We adopt a Deformable DETR-like detection head that outputs the probability of object classes as well as the 3D detection boxes directly without the need for NMS post-processing. The total loss comprises two optimization terms: the mask loss $\mathcal{L}_{mask}$ and the detection loss $\mathcal{L}_{detection}$, as shown in Eq. 8. The mask loss $\mathcal{L}_{mask}$ aims to optimize foreground radar binary classification in the mask module, and the detection loss $\mathcal{L}_{detection}$ optimizes for the 3D detection head, respectively.

$$\mathcal{L}_{total} = \mathcal{L}_{mask} + \mathcal{L}_{detection} \quad (8)$$

Considering the scarcity of foreground radar features, we employ focal loss for training in the foreground radar mask module. For fair comparison with BEVFormer, we adopt focal loss for classification and L1 loss for box regression in the 3D detection head.

## 4 Experiments

### 4.1 Experimental Settings

*4.1.1* **Dataset and Metrics.** We evaluate our approach on the large-scale autonomous driving dataset, nuScenes [1]. It is divided into 700/150/150 scenes for training, validation, and testing, respectively. It provides a 360-degree panoramic view and includes data from multiple sensors: six cameras, one LiDAR, and five radars. For the nuScenes detection task, mean Average Precision (mAP) and NuScenes Detection Score (NDS) are used as the metrics. The mAP, calculated based on the BEV center distance between the predictions and ground truth, is averaged over threshold distances of 0.5, 1, 2, and 4 meters. The NDS is a weighted average of the mAP and additional true positive metrics, which include the Mean Average Translation Error (mATE), Mean Average Scale Error (mASE), Mean Average Orientation Error (mAOE), Mean Average Velocity Error (mAVE), and Mean Average Attribute Error (mAAE).

*4.1.2* **Implementation Details.** By default, in the camera branch, we use ResNet101 [5] pre-trained with FCOS3D as backbone. For neck, we utilize a standard FPN [17], featuring a dimension of 256 and scales of 1/16, 1/32, and 1/64. In Sparse Cross-Attention Encoder, six fusion encoding layers are employed. For the radar branch, we accumulate five previous radar sweeps. Similar to BEVFormer-Base [15], unless specified otherwise, we use three frames of historical BEV features in our Temporal Self-Attention module.

Our model is trained end-to-end for 24 epochs on 8 A100 GPUs using the AdamW [21] optimizer, with learning rate of 2e-4 and batch size of 1. No class balancing strategy (CBGS) [43] or BEV data augmentation [14] is utilized during training.

## 4.2 Comparison with State-of-the-Arts

To evaluate the performance of our SparseInteraction, we conducted comparison using nuScenes [1] val and test set, with results presented in Table 1 and Table 2. Notably, our SparseInteraction surpasses previous state-of-the-art radar-camera fusion methods, achieving 51.1% mAP and 59.5% NDS on the validation set, and 61.7% mAP and 53.8% NDS on the test set without LiDAR auxiliary supervision. Compared with our baseline BEVFormer [15], SparseInteraction shows substantial improvements, achieving **7.8%** and **9.5%** performance gains in NDS and mAP, respectively. These superior metrics underscore the effectiveness of our method.

Significantly, unlike CRN [11], our method does not require extra depth supervision from LiDAR. Although CRN utilizes a more powerful backbone, ConvNextB [18], and incorporates LiDAR data during the training process. Our SparseInteraction still achieves

| Method | Input | Backbone | LiDAR | NDS↑ | mAP↑ | mATE↓ | mASE↓ | mAOE↓ | mAVE↓ | mAAE↓ |
|---|---|---|---|---|---|---|---|---|---|---|
| BEVDepth[†] [14] | C | R101 | ✓ | 53.5 | 41.2 | 0.565 | 0.266 | 0.358 | 0.331 | 0.190 |
| CRN [11] | C+R | R101 | ✓ | 59.2 | 52.5 | 0.460 | 0.273 | 0.443 | 0.352 | 0.180 |
| FCOS3D [34] | C | R101 | | 41.5 | 34.3 | 0.725 | 0.263 | 0.422 | 1.292 | **0.153** |
| DETR3D [35] | C | R101 | | 42.5 | 34.6 | 0.773 | 0.268 | 0.383 | 0.842 | 0.216 |
| BEVFormer-S [15] | C | R101 | | 44.8 | 37.5 | 0.725 | 0.272 | 0.391 | 0.802 | 0.200 |
| CenterFusion[†] [23] | C+R | DLA34 | | 45.3 | 33.2 | 0.649 | 0.263 | 0.535 | 0.540 | 0.142 |
| MVFusion[†] [36] | C+R | R101 | | 45.5 | 38.0 | 0.675 | **0.258** | 0.372 | 0.833 | 0.196 |
| RCBEV[†] [42] | C+R | Swin-T | | 49.7 | 38.1 | **0.526** | 0.272 | 0.445 | 0.465 | 0.185 |
| CRAFT[†] [10] | C+R | DLA34 | | 51.7 | 41.1 | 0.494 | 0.276 | 0.454 | 0.486 | 0.176 |
| BEVFormer [15] | C | R101 | | 51.7 | 41.6 | 0.673 | 0.274 | 0.372 | 0.394 | 0.198 |
| **SparseInteraction-S** | C+R | R101 | | 54.8 | 45.8 | 0.572 | 0.268 | 0.396 | 0.379 | 0.189 |
| **SparseInteraction** | C+R | R101 | | **59.5** | **51.1** | 0.535 | 0.272 | **0.361** | **0.254** | 0.179 |

**Table 1: Performance comparison on the nuScenes val set. "C" indicates Camera, "C+R" indicates Camera and Radar. "SparseInteraction-S" does not leverage temporal information. "LiDAR": using extra LiDAR data source as depth supervision. [†]: trained with CBGS.**

| Method | Input | Backbone | LiDAR | NDS↑ | mAP↑ | mATE↓ | mASE↓ | mAOE↓ | mAVE↓ | mAAE↓ |
|---|---|---|---|---|---|---|---|---|---|---|
| BEVDepth [14] | C | V2-99 | ✓ | 60.0 | 50.3 | 0.445 | 0.245 | 0.378 | 0.320 | 0.126 |
| BEVStereo [13] | C | V2-99 | ✓ | 61.0 | 52.5 | 0.431 | 0.246 | 0.358 | 0.357 | 0.138 |
| CRN [11] | R+C | ConvNextB | ✓ | 62.4 | 57.5 | 0.416 | 0.264 | 0.456 | 0.365 | 0.201 |
| FCOS3D [34] | C | V2-99 | | 42.8 | 35.8 | 0.690 | **0.249** | 0.452 | 1.434 | 0.124 |
| CenterFusion [23] | C+R | DLA34 | | 45.3 | 33.2 | 0.649 | 0.263 | 0.535 | 0.540 | 0.142 |
| DETR3D [35] | C | R101 | | 47.9 | 41.2 | 0.641 | 0.255 | 0.394 | 0.845 | 0.133 |
| RCBEV [42] | C+R | Swin-T | | 48.6 | 40.6 | 0.484 | 0.257 | 0.587 | 0.702 | 0.140 |
| MVFusion [36] | C+R | V2-99 | | 51.7 | 45.3 | 0.569 | 0.246 | 0.379 | 0.781 | 0.128 |
| CRAFT [10] | C+R | DLA34 | | 52.3 | 41.1 | **0.467** | 0.268 | 0.456 | 0.519 | **0.114** |
| BEVFormer [15] | C | V2-99 | | 56.9 | 48.1 | 0.582 | 0.256 | 0.375 | 0.378 | 0.126 |
| RCM-Fusion [9] | C | R101 | | 58.0 | 49.3 | 0.485 | 0.255 | 0.386 | 0.421 | 0.115 |
| **SparseInteraction** | C+R | V2-99 | | **61.7** | **53.8** | 0.497 | 0.254 | 0.375 | **0.269** | 0.121 |

**Table 2: Performance comparison on the nuScenes test set. "C" indicates Camera, "C+R" indicates Camera and Radar. "LiDAR": using extra LiDAR data source as depth supervision. V2-99 is pre-trained on external depth dataset DDAD [4].**

competitive results in terms of NDS and mAVE., leveraging the lightweight backbone, V2-99 [4].

## 4.3 Ablation Study

In this section, we present a comprehensive validation of our framework. For our ablation studies, we utilize a smaller version of SparseInteraction, which includes reducing image resolution from 900x1600 to 700x1260, decreasing frames of historical BEV features from 3 to 2 in our Temporal Self-Attention module, reducing Fusion-BEV queries from 200x200 to 150x150, reducing fusion encoding layers from 6 to 3, and employing R101 as the backbone with image features downsampled by a factor of 32. As shown in Table 3, we start with BEVFormer-Small [15] as our baseline (in #1) and incrementally incorporate each module to assess its impact.

*4.3.1 Noisy Radar Filter.* Comparing #2 and #3 in Table 3 shows the Noisy Radar Filter (NRF) gives **1.1** and **1.5** points improvements on mAP and NDS, respectively. The NRF is composed of

| # | SCAE | NRF | FPQ | NDS[%]↑ | mAP[%]↑ |
|---|---|---|---|---|---|
| 1 | | | | 47.87 | 37.00 |
| 2 | ✓ | | | 54.04 | 44.41 |
| 3 | ✓ | ✓ | | 55.53 | 45.20 |
| 4 | ✓ | ✓ | ✓ | **56.31** (+8.44) | **46.46** (+9.46) |

**Table 3: Ablation of our components on nuScenes val set. BEVFormer [15] is employed as the baseline, and we add the Sparse Cross-Attention Encoder (SCAE), Noisy Radar Filter (NRF) and Foreground Prior Queries (FPQ) in order.**

a *Semantic-enhanced Radar Encoder* and *Foreground Radar Mask*. Various experiments are conducted to evaluate each sub-module within the NRF. In Table 4, setting 1 presents the result with the SCAE module on the baseline, which is the same as #2 in Table 3. Setting 2 illustrates the Semantic-enhanced Radar Encoder effectively bridges the gap on the multi-modal features. However, the

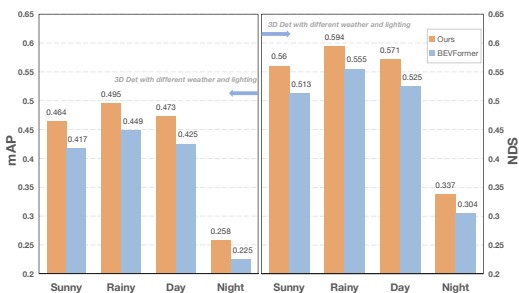

Figure 4: Analysis of robustness under different weather and lighting conditions. Comparing with baseline model, our SparseInteraction achieves superior robustness and performance in all conditions.

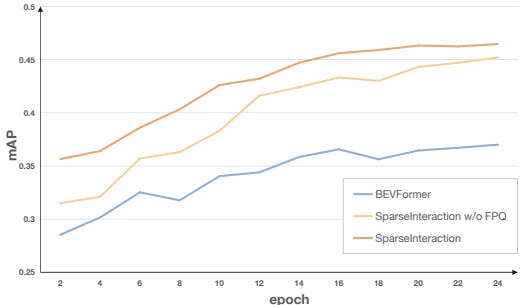

Figure 5: Analysis of convergence using mAP metric. Our SparseInteraction converges ahead of schedule, approximately around epoch 16, while also achieving better detection performance.

| # Setting | SRE | FRM | NDS[%]↑ | mAP[%]↑ |
|---|---|---|---|---|
| 1 | | | 54.04 | 44.41 |
| 2 | ✓ | | 54.30 | 44.82 |
| 3 | | ✓ | 53.62 | 43.75 |
| 4 | ✓ | ✓ | **55.53** | **45.20** |

Table 4: Ablation of Noisy Radar Filter. SRE and FRM denotes Semantic-enhanced Radar Encoder and Foreground Radar Mask, respectively.

| # Fusion Method | NDS[%]↑ | mAP[%]↑ |
|---|---|---|
| Add | 55.06 | 45.11 |
| Concate | 55.39 | 45.68 |
| MFA | 55.45 | 45.83 |
| Ours | **56.31** | **46.46** |

Table 6: Effect of Sparse Cross-Attention Encoder. Note that MFA is a fusion method mentioned in CRN [11].

| # Method | Input | NDS[%]↑ | mAP[%]↑ |
|---|---|---|---|
| BEVDepth | C | 47.49 | 35.94 |
| + Fusion radar | C+R | 51.38 | 38.87 |
| + NRF | C+R | **52.50** | **39.78** |

Table 5: Effect of Noisy Radar Filter module.

| # Method | NDS[%]↑ | mAP[%]↑ | mATE↓ |
|---|---|---|---|
| statistics-based | 56.18 | 45.93 | 0.580 |
| learning-based | **56.31** | **46.46** | **0.569** |

Table 7: Effect of different initial positioning heights in Foreground Prior Queries.

result worsens when extracting the radar foreground features directly using radar information without the semantic context from the image. Ultimately, the Noisy Radar Filter achieves impressive enhancement across all settings, the results as shown in setting 4. To further validate the robustness and versatility of our method, we also conducted experiments on BEVDepth [14]. The results demonstrate consistent benefits, with an increase of **1.1%** mAP and **1.2%** NDS, shown in Table 5, this also highlights the effectiveness of removing noisy radar.

*4.3.2 Sparse Cross-Attention Encoder.* Incorporating the Sparse Cross-Attention Encoder (SCAE) yields an improvement of 1.0% in mAP and 1.2% in NDS presented in Table 3. We compare various fusion strategies including addition, concatenation, and the Multimodal Feature Aggregation (MFA) method mentioned in CRN [11] to demonstrate the effect of our SCAE in Table 6. SCAE aggregates features by directly sampling image features and radar features through FusionBEV Queries, which results in superior fusion features.

*4.3.3 Foreground Prior Queries.* Adding the Foreground Prior Queries (FPQ) contributes a gain of 1.0% mAP and 1.2% NDS as

shown in Table 3. To evaluate the effect of initial positioning height in FPQ, we conducted experiments comparing the learning-based method with statistics-based approaches, shown in Table 7. The statistics-based refers to use the average height of ground truth bounding boxes in the dataset as the initial height for FPQ. In contrast, the learning-based method acquires different heights on the BEV plane. Unlike the fixed value in the statistics-based approach, this method simplifies the learning complexity for object queries at heights, thereby enhancing performance.

## 4.4 Analysis

*4.4.1 Weather and Lighting.* We analyze the robustness and performance under different weather and lighting conditions in Figure 4. It is noted that detection during nighttime poses challenges for camera-only method. Thanks to the stable performance of radar under adverse weather conditions, our fusion model achieves consistent improvements in sunny and rainy weather, as well as daytime and nighttime conditions, compared with baseline model. This demonstrates the superior robustness and performance of our proposed SparseInteraction in all conditions.

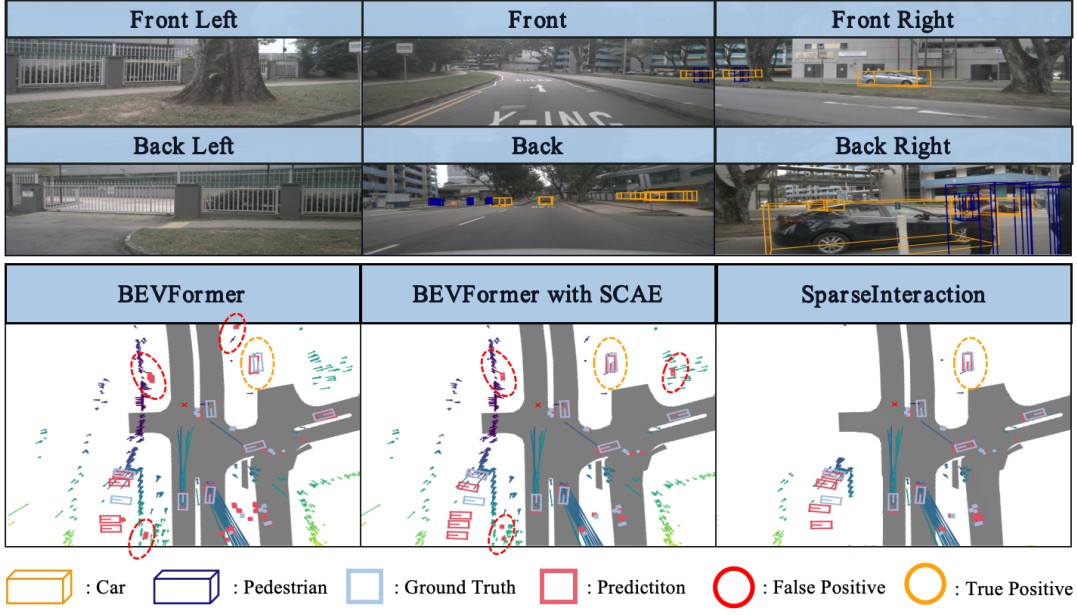

**Figure 6: Visualization of predictions from different models. To validate the effectiveness of the sub-module in SparseInteraction, we visualize each step's result on our baseline BEVFormer. Specifically, SCAE represents the proposed fusion module, the Sparse Cross-Attention Encoder. The first line illustrates framework predictions in the image view, while the second line showcases detection results in BEV space. SparseInteraction effectively eliminates noisy radar inputs and enhances fusion features, leading to a notable reduction in false positives.**

*4.4.2* **Convergence.** We perform the analysis on the convergence of SparseInteraction in Figure 5. Thanks to the aid of radar, Sparse Interaction without FPQ consistently achieves significantly better detection performance compared to BEVFormer. However, their rates of convergence are identical, which we believe is attributed to them utilizing the same decoder head. Furthermore, because FPQ provides prior location information, it enables SparseInteraction to converge ahead of schedule, approximately around epoch 16, while also enhancing detection performance.

*4.4.3* **Performance and Latency.** We compare the performance and latency using different frames in Table 8. After observing that the utilization of multiple temporal frames significantly enhances NDS, mAP, and mAVE while the best performance is achieved with four frames. Consequently, we opted to use four frames, taking into consideration computation time and memory constraints during training. Since we pre-save the history fusionBEV features during testing, the use of temporal information does not increase latency. Furthermore, in the Multi-Modal Cross Attention module, the exclusion of FusionBEV queries at background positions from computations enables the FusionBEV queries to focus on potential target object positions. Consequently, our model achieves lower latency and better performance compared to baseline model.

## 4.5 Visualization

We provide a more straightforward visualization in Figure 6. Comparing the predictions of BEVFormer, we note that enhanced fusion

| # Frame | # Method | NDS↑ | mAP↑ | mAVE↓ | FPS |
|---------|----------|------|------|-------|-----|
| 4 | BEVFormer | 0.517 | 0.416 | 0.394 | 1.7 |
| 1 | | 0.548 | 0.458 | 0.379 | **2.1** |
| 2 | | 0.564 | 0.483 | 0.326 | **2.1** |
| 3 | SparseInteraction | 0.586 | 0.505 | 0.283 | **2.1** |
| 4 | | **0.595** | **0.511** | 0.254 | **2.1** |
| 5 | | **0.595** | 0.507 | **0.247** | **2.1** |

**Table 8: Analysis of performance and latency using different frames during training. "# Frame" denotes the frame number during training.**

features can refine the bounding box by SCAE. However, this process is sometimes disrupted by noisy radar, leading to false positives. In contrast, when comparing BEVFormer to SparseInteraction, it becomes evident that our framework successfully eliminates noisy radar data before fusion, effectively preventing false positives.

## 5 Conclusion

In this paper, we proposed a novel transformer-based foreground radar camera fusion framework with a light module that extracts foreground radar features under the guidance of semantic image. Our approach effectively addresses the inherent challenges of noise and positional ambiguity which achieves state-of-the-art performance. In future work, we plan to incorporate velocity prediction into the temporal module to further enhance the detection performance.

# Acknowledgments

This work was supported in part by the Science and Technology Development Fund, Macau SAR, under Grant 0075/2023/AMJ, Grant 0003/2023/RIB1, and Grant 001/2024/SKL; in part by the Guangdong Science and Technology Department under Grant 2023A0505030003 and Grant 2020B1515130001; in part by the Zhuhai Science and Technology Innovation Bureau under Grant ZH2220004002524; in part by the International Science and Technology Project of Guangzhou Development District under Grant 2022GH09; in part by the Zhuhai UM Research Institute under Grant HF-011-2021; and in part by the University of Macau under Grant MYRG2022-00059-FST and Grant MYRG-GRG2023-00237- FST-UMDF.

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
