# OpenReview forum: "SparseInteraction: Sparse Semantic Guidance for Radar and Camera 3D Object Detection"
_acmmm.org/ACMMM/2024/Conference — MM2024 Poster_

### Official Review · Reviewer_BmR3 · 2024-05-21

**Rating:** 5
**Confidence:** 1

**Summary:**

The paper introduces SparseInteraction, an end-to-end fusion framework, to enhance detection performance. It employs the Noisy Radar Filter (NRF) module to extract foreground features by filtering out noisy radar data using semantic image features. The Sparse Cross-Attention Encoder (SCAE) effectively combines foreground radar and image features to resolve positional ambiguities. To further improve model performance and convergence, foreground prior queries with positional information are integrated into the transformer-based decoder. Experimental results show that these fusion strategies achieve state-of-the-art results on the nuScenes benchmark.

**Strengths:**

1. A novel Noisy Radar Filter was proposed to efficiently address false positive issues.
2. Sufficient experiments on nuScenes dataset demonstrate the superiority of the proposed method.

**Limitations:**

1.Is Eq.1 the first time proposed? Please provide more references to highlight your contribution

2.Foreground Radar Mask seems like an interesting concept. Is there any previous work proposing similar ideas? Could it be visualized in a paper?

3.Some recent open-source works can be compared in the experiment.
[1] Wang S, Liu Y, Wang T, et al. Exploring object-centric temporal modeling for efficient multi-view 3d object detection[C]//Proceedings of the IEEE/CVF International Conference on Computer Vision. 2023: 3621-3631

4.Some recent works can be introduced in the related works.
[1] Jiang X, Li S, Liu Y, et al. Far3d: Expanding the horizon for surround-view 3d object detection[C]//Proceedings of the AAAI Conference on Artificial Intelligence. 2024, 38(3): 2561-2569.
[2] Yang C, Chen Y, Tian H, et al. BEVFormer v2: Adapting modern image backbones to bird's-eye-view recognition via perspective supervision[C]//Proceedings of the IEEE/CVF Conference on Computer Vision and Pattern Recognition. 2023: 17830-17839.

**Suitability:**

3

---

### Official Review · Reviewer_EX3z · 2024-05-24

**Rating:** 5
**Confidence:** 4

**Summary:**

The paper presents a framework called SparseInteraction , designed for 3D object detection using radar and camera data. This multi-modal fusion approach addresses the limitations of existing methods by mitigating noise and positional ambiguity inherent in radar data. The framework includes key components such as the Noisy Radar Filter (NRF) and Sparse Cross-Attention Encoder (SCAE), which enhance radar features using semantic information from images and blend these features effectively to improve detection performance. The approach is validated through experiments on the nuScenes benchmark, showing state-of-the-art results.

**Strengths:**

Firstly, the proposed NRF module effectively filters out noisy radar data by leveraging semantic information from images. This reduces false positives and enhances the overall quality of the radar features used for fusion, leading to more accurate detection.

Secondly, the Sparse Cross-Attention Encoder (SCAE) accurately fuses radar and image features by addressing radar data's positional ambiguity. This fusion method allows for precise feature alignment and interaction, boosting overall detection performance.

Thirdly, SparseInteraction achieves high detection performance on the nuScenes dataset without relying on LiDAR data for depth supervision. It excels in metrics such as mean Average Precision (mAP) and NuScenes Detection Score (NDS), outperforming existing radar-camera fusion methods.

**Limitations:**

1.The proposed method relies on extracting high-precision semantic information from images. However, its effectiveness in maintaining high performance under conditions of poor image quality, such as adverse weather or low lighting, is uncertain and requires further validation.
2.The experiments in the paper are solely conducted on the nuScenes dataset. To robustly verify the effectiveness of the proposed method, additional comparisons on other mainstream 3D detection datasets like Waymo and KITTI are necessary.
3.The paper would benefit from including detailed visualizations of 3D detection results in the ablation studies. This would allow for a thorough analysis of the degradation effects caused by the absence of specific components, providing deeper insights into the contributions of each part of the model.

**Suitability:**

2

---

### Official Review · Reviewer_nBPK · 2024-05-24

**Rating:** 3
**Confidence:** 2

**Summary:**

This paper introduces an end-to-end fusion framework that enhances radar and camera 3D object detection by using a Noisy Radar Filter (NRF) to filter out noisy radar data with semantic image features and a Sparse Cross-Attention Encoder (SCAE) to efficiently merge these features, achieving state-of-the-art performance without requiring LiDAR supervision.

This approach significantly improves detection accuracy and robustness, especially in adverse weather and lighting conditions, surpassing existing methods on the nuScenes benchmark.

**Strengths:**

1. This paper enhances 3D object detection by introducing an innovative Noisy Radar Filter (NRF) module that uses semantic features from images to filter out noisy radar data, and an efficient Sparse Cross-Attention Encoder (SCAE) that merges radar and image features at a sparse level, improving precision and reducing computational complexity.
2. This paper effectively addresses the challenge of 3D object detection without relying on LiDAR data, utilizing only camera and radar inputs.

**Limitations:**

1. This paper uses different backbones for evaluations on the validation set (ResNet101) and the test set (V2-99), which may affect the fairness and consistency of performance comparisons.
2. This paper only compares its results with RCM-Fusion’s camera-only (C) results, not its camera and radar (C+R) results. RCM-Fusion uses a single-frame approach, whereas SparseInteraction employs a multi-frame approach. Additionally, RCM-Fusion uses a multi-modal fusion pipeline, which could weaken the claim of focusing on the characteristics of a single modality before filtering.

**Suitability:**

2

---

### Official Review · Reviewer_SfHm · 2024-06-08

**Rating:** 3
**Confidence:** 2

**Summary:**

This paper presents a robust end-to-end fusion framework that effectively filters noisy radar data using semantic features from images and integrates these with radar data using a Sparse Cross-Attention Encoder, achieving state-of-the-art performance on the nuScenes benchmark.

**Strengths:**

A new fusion strategy significantly improves detection performance and robustness across various conditions, demonstrating superior results compared to existing methods without relying on LiDAR data.

**Limitations:**

1.How does the proposed Noisy Radar Filter (NRF) quantitatively compare to traditional radar filtering techniques in terms of noise reduction and feature enhancement?
2.Can the authors provide more detailed results on the performance of SparseInteraction across different weather and lighting conditions? How consistent are the improvements in these varying scenarios?
3.What is the computational overhead introduced by the NRF and SCAE modules, and how does it impact the overall processing time and feasibility of real-time application in autonomous driving?
4.Has the SparseInteraction framework been tested on datasets other than nuScenes? If not, how do the authors justify the generalization capability of their model across different datasets and real-world conditions?
5.The paper mentions ablation studies for various components. Can the authors provide more detailed insights into the individual contributions of each component (NRF, SCAE, FPQ) to the overall performance improvement?

**Suitability:**

2

---

### Meta-Review · Area_Chair_4UfH · 2024-06-25

**Recommendation:** Accept (Poster)
**Confidence:** 5

**Metareview:**

The paper "SparseInteraction" introduces a novel fusion framework for integrating radar and camera data to improve 3D object detection. The key contributions include the Noisy Radar Filter (NRF) module, which utilizes semantic features from images to filter out noisy radar data, and the Sparse Cross-Attention Encoder (SCAE) that merges radar and image features to address positional ambiguities. The method achieves state-of-the-art performance on the nuScenes benchmark, demonstrating significant improvements in detection accuracy and robustness under various conditions.

While the reviewers have highlighted some limitations, such as the need for further validation on additional datasets and scenarios, the overall consensus leans towards the strength and innovation of the proposed framework. The authors have addressed most concerns in their rebuttal, showcasing the effectiveness and potential of SparseInteraction in advancing radar-camera fusion techniques.

Given the substantial performance improvements and the novelty of the approach, the paper makes a contribution to the field of multimodal perception and 3D object detection.